# Impact of Caregiving for Dementia Patients on Healthcare Utilization of Caregivers

**DOI:** 10.3390/pharmacy7040138

**Published:** 2019-09-24

**Authors:** Ateequr Rahman, Rubeena Anjum, Yelena Sahakian

**Affiliations:** 1College of Pharmacy, Rosalind Franklin University of Medicine and Science, North Chicago, IL 60064, USA; yelena.sahakian@my.rfums.org; 2Silver Cross Hospital, New Lenox, IL 60451, USA

**Keywords:** caregivers, dementia patients, healthcare utilization, healthcare services

## Abstract

The elderly, whom are vulnerable to the physical, mental and chronic diseases of aging, are the fastest growing segment of the US population. Dementia is of particular concern in this population, and caregivers of people with dementia are subjected to psychological, physical, emotional and functional stress. The purpose of this study was to investigate the impact of caregiving for dementia patients on health care services utilization of caregivers and to examine if caregivers utilize more healthcare services than the control group. The study recruited a total of 143 people in control and non-control groups through non-probability convenience sampling. The control group (non-caregivers) comprised of 71 people, whereas the experimental group (caregivers) consisted of 72 participants. The focus of the study was the health care utilization questionnaire, asking the caregiver about the frequency of specific health care services utilization—including medication use in the last six months, on the scale from 0 to 10. Results were statistically significant for each of the healthcare service utilization when comparing caregivers to the control group. By providing adequate support and assistance in form of support groups, we can alleviate caregivers’ burden and more effectively address the needs of caregivers—thereby reducing the utilization of healthcare services.

## 1. Introduction

Around 18 million people worldwide have already been diagnosed with Alzheimer’s disease, according to the World Health Organization [1,2]. Dementia of the Alzheimer’s type is projected to increase four-fold by 2050 globally due to declining birth, mortality rates, and increased life-extending medical care. People whom are 65 years and older are projected to represent 20% of the population by 2030. Those that are 75 years and older comprise the fastest growing segment of the population and are the most vulnerable to both physical and mental diseases [3].

Alzheimer’s disease manifests itself as a consistent and progressive deterioration of memory. People with Alzheimer’s disease experience a gradual and progressive decline in short-term memory and activities of daily living. This can cause social and occupational embarrassment and eventually lead to dementia characterized by impaired cognition, impaired daily activities and an increase in functional dependence [4].

Caregivers, defined as families and friends of people with dementia, are a primary source of home care and support—especially when this is a cultural expectation. They contribute to services that would otherwise cost hundreds of billions of dollars annually if they had to be purchased. In addition, caring for a person with dementia is typically more challenging than caring for a person with physical disabilities alone.

Caregiving is a dynamic process characterized by changes in sleep and stress that contribute to changes in caregiver mood and health [5]. It has been shown that caregivers of people with dementia often have higher levels of stress and depression compared to caregivers of older adults who have suffered from other health issues. Overall, stress associated with caregiving leads to lower self-ratings of health. A study of about 19,000 American Association of Retired Persons (AARP) Medicare Supplement insurance carriers, caregivers that live with the impaired person, reported loneliness. The results indicated that caregiving negatively impacted their psychological health [6].

In a meta-analysis, it was indicated that when compared to non-caregivers, caregivers had a prevalence of chronic illnesses, such as hypertension, arthritis and heart disease [7]. Female caregivers were at a higher risk for depression and stress than their male counterparts, as well as older caregivers when compared to younger caregivers [2,8]. A study reported that caregivers of people with dementia had a higher consumption of drugs and nonpharmacological therapies, such as psychologists, therapists, social workers and support of an association of patient relatives. They consumed more anxiolytics, antidepressants, and antiplatelet medications than the control group. They found as well that nonpharmacological therapies were used significantly more in the caregivers’ group than in the control group [9].

Some studies have compared health care resources utilization between caregivers and non-caregivers. In an 18-month-long study, the caregivers of Alzheimer’s patients self-reported their health declining steadily and significantly. Emergency room visits and hospital-based services doubled over time, showing an overall 25% trend in increased use of all types of health services. Caregivers who rated their health as “fair” or “poor” were more likely to utilize healthcare resources, which was calculated to be US $4766 on average annually [10].

Caregiving burden threatens the psychological, physical, emotional and functional health of caregivers. They report more physical and psychological symptoms and use more frequent prescription medications and healthcare services than comparable non-caregivers [11].

Caregivers face many obstacles as they balance caregiving with other family responsibilities, personal relationships and their career. Almost 60% of US family caregivers of people with dementia are employed as well, of whom 2/3 reported that they missed work, 8% turned down promotion opportunities, and up to 31% had given up work to attend to caregiving responsibilities [5]. National Health and Wellness Survey (NHWS) data analysis showed that caregivers report a higher mean percentage of work time missed (8% versus 4%, *P* < 0.05) and greater productivity impairment (24% versus 14%, *P* < 0.05) [12].

A secondary database analysis of NHWS reported that caregivers had a higher prevalence of smoking (26% versus 19%, *P* < 0.05) and insomnia (46% versus 37%, *P* < 0.05) [12].

The role of dementia care networks is valuable for caregivers. Dementia care networks are community-based collaborative organizations providing dementia care services to local communities. The networks bring together nonprofit human services providers, program consumers, community representatives and government entities according to the needs of their target populations [13]. More formalized knowledge management in dementia care networks can lead to more knowledge among family caregivers. Overall, such support networks increase the quality of information available and improve support for people with dementia and their caregivers [14].

A number of studies have shown that teaching caregivers coping skills may help build some resilience. Providers of dementia patients can teach their caregivers techniques of coping with anger, depression and frustration, and thus improve the caregiver’s quality of life [15]. A systematic review of 10 meta-analyses has suggested the effectiveness of professional self-management support interventions—defined as support from providers or social groups to improve an individual’s ability to manage the symptoms, treatment, physical and psychosocial consequences and lifestyle changes due to caring for a person with a chronic condition. They are aimed to reduce stress, improve psychological wellbeing and the social outcomes [16].

The purpose of this study was to explore the utilization of healthcare services by caregivers of dementia patients provided at home as compared to non-caregivers.

## 2. Methods

Caregiving requires a great deal of time, patience and can cause great stress to the caregiver. This study hypothesized that caregivers for people with dementia have greater health care needs and therefore utilize more healthcare services as compared to non-caregivers.

The study recruited a total of 143 people in control and non-control groups through non-probability convenience sampling. The control group (non-caregivers)—who did not provide care and did not meet caregiver criteria—comprised of 71 people, whereas the experimental group (caregivers) consisted of 72 participants.

Caregivers were operationally defined as people sharing a single household with a person with dementia for the past year, not receiving financial compensations and bearing the primary responsibility for the impaired person. Another inclusion criterion were evident symptoms of dementia, memory loss and disorientation of the person with dementia. Non-caregivers were the participants who did not provide care in their household and did not meet caregivers’ criteria.

After obtaining permission from various local community centers, churches, mosques and temples in Illinois suburbs, the researchers visited the facilities on their weekly service days. The surveys were distributed to the participants of the daily services. The choice of locations was based on vicinity and convenience of obtaining approvals.

An announcement and a brief introduction about the study was given to the audience. Rights of all participants were protected. The researchers first introduced themselves, spoke about the importance and the relevance of the study, and solicited participation by addressing the gatherings. The research was conducted according to the principles of the *Declaration of Helsinki*. A cover letter and a consent form accompanying the questionnaire stated that all responses would be treated confidentially; the participants were not identified with their responses and had the right to withdraw from the study at any time.

Both groups received the same survey instrument; the difference being that one group was a caregiver—whereas the other one was not. The questionnaire first asked the caregiver’s relationship to the person with dementia or Alzheimer’s disease. The survey instrument used for data collection had demographic (16 questions) and health care services utilization (20 questions) parts. The demographic section included questions on gender, ethnicity, marital status, monthly income and employment status—self-identified by the participants. The health care utilization portion of the survey consisted of questions about the frequency of visits to various healthcare services and support groups. The 10-point scale was used to determine how many times the caregiver utilized healthcare services in the past six months, with 0 being not using any services and 10 being utilizing such services 10 or more times in the last six months. Other questions in the same section asked whether the caregiver had health insurance or changes in health status. The survey specified the visits to the type of health care provider and healthcare services used by the caregiver. Incomplete or partially finished surveys were not included in analysis.

Using responses based on Likert scale, the participants completed a self-administered questionnaire designed to test the research objectives. The ambiguity of the questionnaire was assessed in terms of flow and the respondents’ comprehension of the words used in the questionnaire. An apriori level of 80% or greater consensus among the respondents was set and accepted. The reading comprehension difficulty of the questionnaire was measured in terms of the Flesch reading score and Flesch-Kincaid grade level. An apriori grade level of 6 or less was set. The Flesch reading ease score was found to be 65.2 and the Flesch-Kincaid grade level was found to be 6.9. The Likert-type scale was tested for reliability on the pretest and posttest using Cronbach’s coefficient alpha. Alpha was 0.93 for the pretest group and 0.95 for the posttest group; most educational tests have a reliability of 0.40 to 0.95 [17].

A series of descriptive analyses such as percentages, mean and standard deviations were performed on the demographic variables. The Chi square test and t-test were used to assess and compare the health care utilization of caregivers with the control group. Since the data was categorical in nature, Chi square was the most appropriate test for analyzing the independent variables. The Chi square (*X*^2^) statistics were used to investigate whether distributions of categorical variables differed from one another.

The independent variables were compared with the dependent variables such as various resource utilizations to explore possible relationships by using t-tests. Independent samples of the t-test compare the means of two independent groups in order to determine whether there is statistical evidence about associated population means being significantly different.

A probability level of 0.05 was used for all analyses. The data were analyzed using Statistical Package for Social Sciences (SPSS^R^-25) program.

## 3. Results

The study found that the majority of the caregivers and controls were female (69%). The largest ethnic group represented were Asians (56%), with both Caucasian (31%) and Hispanic (13%) ethnic groups being present as well. The surveys were distributed at the predominantly Asian facilities due to convenience and ability to obtain a large number of participants at a given time at those facilities. This explains the large percentage of Asians in the sample size (Table 1).

The mean age of the participants was homogenous between the two groups with 47 and 52 years old for the control and the caregiver group, respectively. The majority of the caregivers in the sample were parent/grandparent, followed by friend and spouse (Table 2). More than half of the caregivers in the experimental group were married.

Most participants were high school graduates (22%) in the caregivers group and (28%) in the control group; college graduates (68%) in the caregivers group and (60%) in the control group; with the majority having a monthly income of over $3500 per month with (64%) in the caregivers group and (72%) in the control group.

The focus of the study was the health care utilization questionnaire, asking the caregiver “since you have started caregiving, has your use of health care services (including medication consumption): increased, decreased or remained the same.” The 10-point scale was used to determine how many times the caregiver utilized healthcare services in the past six months. The 10-point scale was graded from “0”, meaning no service was utilized, to “10+”, meaning the service was utilized more than 10 times over the six-month span. Results were statistically significant for each of the healthcare service utilization when comparing caregivers to the control group.

All ethnicities stated that their health care service utilization has increased since caregiving. The findings were statistically significant (*P* < 0.05), with 94% of Asians, 100% of Hispanics, and 60% of Caucasians reporting increased healthcare utilization (Table 1).

The overall responses to the survey showed statistically significant difference (*P* < 0.05) between the caregiving group and the control group in their utilization of health care services. This indicates that caregivers in fact utilize more healthcare services than non-caregivers (Figure 1).

## 4. Discussion

Utilization of health care services was significantly higher for caregivers in every category (*P* < 0.05) compared to non-caregivers. Visits to general practitioners were three times more frequent than those in the control group. This trend was consistent throughout all health care services. Visits to pharmacists increased twice, and psychologist visits increased three times. Therefore, it is of the utmost importance that interventions for caregivers providing care in home health are designed to help specific groups of caregivers that are at greater risk of developing these symptoms. The frequency of visits to social workers and occupational/physical therapists increased significantly as well as compared to the non-caregiver group as reflected in Figure 1.

The use of over-the-counter (OTC) medications and prescription medications were significantly higher as well compared to that of the control group. A study by the National Alliance for Caregiving and Evercare reported similar trends in terms of increased medication use in caregivers. This study reports that caregivers use prescription and psychotropic drugs more than non-caregivers [18]. The survey only identified the frequency or prescription and OTC medication use rather than the class of medications consumed. Regardless, it can be implied that this finding was related to the physical and psychological burden associated with caregiving. In addition, caregivers utilized more psychological and social services as compared to the non-caregivers. Thus, the current study suggests incorporation of support group meetings for dementia caregivers during regular doctor visits for people with dementia. This will ensure the availability of the resources, such as alzhemiers.gov to look for support group meetings and programs offered at different locations. The use of support groups can significantly reduce the need for psycho-social services, indicating the importance of support group participation and health benefits for the caregiver [16,19].

The findings in the scientific literature regarding whether the health status associated with caregivers of dementia patients is poorer or better, are inconsistent. There is inconsistency of the available literature on healthcare utilization differences among caregivers and non-caregivers as well. Some of the studies show poorer health status [20,21] and behavioral problems [22], while others report caregiving having positive effects on health, provided caregiving activities are not too heavy a burden [23], decreased hospitalizations [24] and even decreased all-cause mortality [25].

The majority in the caregivers group had some type of income. The monthly income amount could be related to their pensions or social security benefits. Since most of the caregivers were homemakers, they got to spend the majority of their time with dementia patients, which could add to their stress, physical and mental exhaustion, and adversely impact their psychological well-being.

Another finding of the study was that females were more likely to provide caregiver services compared to males. This finding is in line with other findings where women are often the primary caregivers [6,26,27]. In some cultures, caring for a sick family member is very natural. In some cases, children of impaired older adults, who have families of their own, jobs and other responsibilities seem to handle the responsibility of intensively taking care of a sick person much better. Culture may have played some role in the fact that the half of the participants in the caregivers’ group were Asians, followed by Caucasians and Hispanics. This is consistent with other studies suggesting that culture, upbringing and individual beliefs are a few determinants of a person’s ability to cope with stress [28,29]. Our data showed as well that the majority of the caregivers were homemakers or retired. Caring for a person with dementia is a 24/7 responsibility, and there is little to no time left for one’s personal life [30].

One limitation of our study was the small sample size—even though the experimental group was matched with a control group of a similar size. Another limitation was that the caregivers were self-identified, and the researchers were not in the position to verify their response. The sample was not randomized, but rather chosen on the basis of convenience sampling; thus, the results cannot be generalized. There was a mismatch of the caregiver and the control groups in terms of employment, income and marital status. Furthermore, the findings might be biased from the recruitment strategy and the possible influence of faith and engagement in faith/social networks. Future studies are recommended to use a larger sample. Our sample was a fairly good representative of the population in terms of including groups from Asian, Caucasian and Hispanic ethnicities. Future studies should focus on the investigation of the use of support groups or developing information seminars and training programs for caregivers with the goal of teaching them coping mechanisms. These intervention or prevention techniques can assist caregivers in maintaining their own health. This preliminary study may lay the groundwork for subsequent studies examining the relationship between various socio-demographic factors and health care utilization of caregivers of people with dementia.

Caregivers of family members with dementia face many health challenges ranging from chronic illnesses to depression and anxiety [19,20,22]. It is important to address this issue at early stages of caring rather than face them in their critical phases. Support groups, such as dementia caregiving networks and visits to psychologists can alleviate the caregivers’ burden and mental exhaustion, which piles up due to lack of sleep, stress and physical depletion. Health problems that dementia caregivers face, translate into more visits to specialists and thus increased healthcare costs. The study found that almost all healthcare services were utilized significantly higher amongst caregivers as compared to the non-caregivers—including psychological, social and clinical services. By accurately measuring and identifying health needs and health care services utilization of caregivers, it is possible to more effectively address the special needs of caregivers of people with dementia.

## Figures and Tables

**Figure 1 pharmacy-07-00138-f001:**
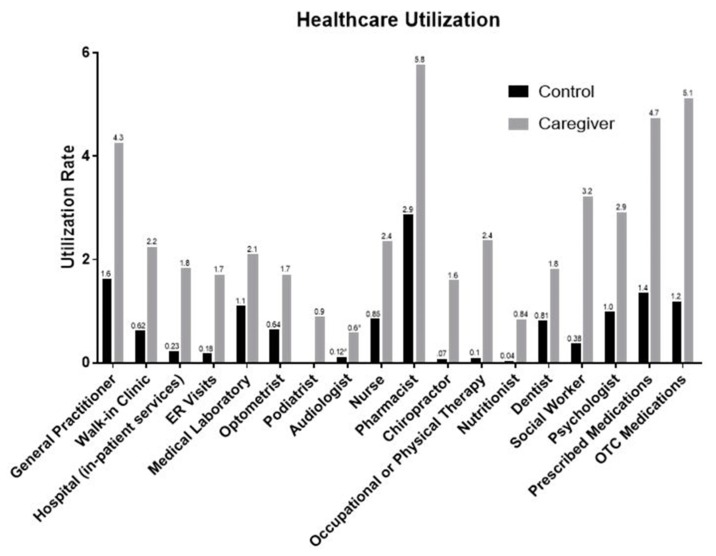
Utilization rate of the healthcare services.

**Table 1 pharmacy-07-00138-t001:** Summary of Demographics.

Demographics	Caregiver n = 71	Control n = 73
Variable	Percentage (%)	Percentage (%)
**Gender**		
Male	18.3	22.8
Female	81.7	77.2
**Ethnicity**		
Caucasian	28.2	34.2
Hispanic	19.7	5.5
Asian	50.7	60.3
**Marital Status**		
Married	54.9	31.5
Widowed	29.6	19.8
Divorced	8.5	11
Single	7	37.7
**Monthly Income**		
$0–$499	0	2.7
$500–$1499	0	16.4
$1500–$2499	7	12.3
$2500–$3499	19.7	13.7
$3500+	73.2	54.8
**Employment Status**		
Retired	23.9	6.8
Part-time	2.8	20.5
Full-time	14.1	60.3
Homemaker	50.7	1.4
Other	0	11

**Table 2 pharmacy-07-00138-t002:** Caregiver relationship to the patient.

Relationship to the Patient		n = 71	
		Frequency (n)	Percentage (%)
**Relationship**	Spouse	19	26.8
	Friend	4	5.6
	In-laws	16	22.5
	Parent/grandparent	32	45.1

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
