# Peer review of "Impact of Caregiving for Dementia Patients on Healthcare Utilization of Caregivers"

_pharmacy, 2019, doi:10.3390/pharmacy7040138_

Round 1

Reviewer 1 Report

Thank you for the opportunity to review your interesting paper which I think has merit but could be improved. I have included an annotated version with details on areas to be addressed. There are several inaccuracies and inconsistencies. The Tables and Figure need legends explaining the content. There is content missing from Table 1. Limitations don't address the differences between study groups. The Discussion needs to be supported by references.The Conclusion needs to more accurately reflect the study aim/hypothesis. Remember to update the Abstract once the main text has been updated. Look forward to seeing your revised manuscript as this is a worthwhile study.

Author Response

RESPONSE TO REVIEWER 1 COMMENTS

Comments on the annotated pdf version of the manuscript

Point 1: reference 7 is incorrect.

Response 1: Reference 7 is in AMA format, just like the rest of the references

Point 2: “Is this the right way round for caregivers and control?”

Response 2: It was a convenience sample. The researchers did not have the option of recruiting equal number of participants in each group, since it was a voluntary participation.

Point 3: Ethics statement towards the end of the Methods.

Response 3: The comment has been addressed and reflected on lines 106-112.  

Point 4 &5: tables and the figure need legends. Misalignment in Table 1.

Response 4&5: Table titles and legends have been added. Misalignment in Table 1 has been fixed.

Point 6: show full extent of scale to 10+ on figure 1.

Response 6: The full extent of the figure up to 10+ is not depicted due to logistic and graphical reasons. None of the values exceed 10, or even 6. The extent of 6 was chosen so the difference between the test and the control groups was zoomed in and easy to notice.

Pointe 7: reference required-line 193.

Response  7: The references have been added. See the manuscript (highlighted in yellow) line 198 on the edited version.

Point 8: line 195 – typo.

Response  8:Typo fixed (line 205 on the edited version).

Point 9: line 195 – “most of them were college graduates”.

Response 9: That particular sentence has been deleted to address the comment.

Point 10: line – 199 – “as well as psychological well-being”

Response 10: Has been edited to “adversely impact to psychological well-being”.

Point 11: reference needed

Response 11: The sentence has been removed. No reference has been added to the sentence before that, since it is a common sense.

Point 12:  mismatch of caregivers and non-caregivers.

Response 12: The limitation has been added to the manuscript (see lines 225-226 on the edited version).

Point 13: This paragraph should more closely address the stated aim of the research/hypothesis and be based in the research study findings.

Response 13: The comment has been addressed. See lines 239-242 on the edited manuscript, which reads as follows: “Health problems that dementia caregivers face, translate into more visits to specialists, thus increased healthcare costs. The study found that almost healthcare services were utilized significantly higher amongst caregivers as compared to the non-caregivers, including psychological, social and clinical services”. 

Point 14: references needed.

Response 14: References have been added. See the manuscript (line 236 on the edited version).

Additional comments

Point 1: There are several inaccuracies and inconsistencies. The Tables and Figure need legends explaining the content. There is content missing from Table 1.

Response 1: The changes have been highlighted in the manuscript.

Point 2: Limitations don't address the differences between study groups.

Response 2: The limitation about the mismatch has been addressed in the manuscript and highlighted in yellow.

Point 3: The Discussion needs to be supported by references.

Response 3: References, citations, as well as comparison with the available data have been added to the manuscript and highlighted in yellow in Discussion section.

Point 4: The Conclusion needs to more accurately reflect the study aim/hypothesis.

Response 4: See lines 239-242 highlighted in yellow in the edited manuscript.

Point 4: Remember to update the Abstract once the main text has been updated.

Response 5: The abstract has been rewritten as follows: “The elderly is the fastest growing segment of the US population and is vulnerable to physical, mental and chronic diseases of aging. Dementia is of particular concern in this population. Caregivers of people with dementia are subjected to psychological, physical, emotional and functional stress. The purpose of this study was to investigate the impact of caregiving for dementia patients on health care services utilization of caregivers and to examine if caregivers utilize more healthcare services than the control group. The study recruited a total of 143 people in control and non-control groups through non-probability convenience sampling. The control group (non-caregivers) comprised of 71 people, whereas the experimental group (caregivers) consisted of 72 participants. The focus of the study was the health care utilization questionnaire, asking the caregiver about the frequency of specific health care services utilization, including medication use, in the last 6 months, on the scale from 0 to 10. Results were statistically significant for each of the healthcare service utilization when comparing caregivers to the control group. By providing adequate support and assistance in form of support groups, we can alleviate caregivers’ burden, more effectively address the needs of caregivers, thereby reducing the utilization of healthcare services”.

Reviewer 2 Report

The abstract should be written in a better way. It has to contain more information about methodology and main results.

A very complicated way of writing your aim. Please rewrite it. "This study hypothesized that caregivers for people with dementia have greater health care needs and therefore utilize more healthcare services as compared to non-caregivers" (lines 90-92).

Can you please give more details about the questionnaire. How many questions, in how many sections. Did you have only a Likert scale type of questions?

Was your sample randomly selected?

Line 212-216 Some of your text should be in the methodology section too. 

In my opinion, you need to expand your discussion based on your results. More comparisons with the current literature should be made. Do not focus only on your limitations. 

Author Response

RESPONSE TO REVIEWER 2 COMMENTS

Point 1: The abstract should be written in a better way. It has to contain more information about methodology and main results.

Response 1: The abstract has been rewritten as follows: “The elderly is the fastest growing segment of the US population and is vulnerable to physical, mental and chronic diseases of aging. Dementia is of particular concern in this population. Caregivers of people with dementia are subjected to psychological, physical, emotional and functional stress. The purpose of this study was to investigate the impact of caregiving for dementia patients on health care services utilization of caregivers and to examine if caregivers utilize more healthcare services than the control group. The study recruited a total of 143 people in control and non-control groups through non-probability convenience sampling. The control group (non-caregivers) comprised of 71 people, whereas the experimental group (caregivers) consisted of 72 participants. The focus of the study was the health care utilization questionnaire, asking the caregiver about the frequency of specific health care services utilization, including medication use, in the last 6 months, on the scale from 0 to 10. Results were statistically significant for each of the healthcare service utilization when comparing caregivers to the control group. By providing adequate support and assistance in form of support groups, we can alleviate caregivers’ burden, more effectively address the needs of caregivers, thereby reducing the utilization of healthcare services”.

Point 2: A very complicated way of writing your aim. Please rewrite it. "This study hypothesized that caregivers for people with dementia have greater health care needs and therefore utilize more healthcare services as compared to non-caregivers" (lines 90-92).

Response 2: See lines 91-92 -The purpose of this study was to investigate whether caregivers of dementia patients utilize more healthcare services than the control group.

Point 3: Can you please give more details about the questionnaire. How many questions, in how many sections.Did you have only a Likert scale type of questions?

Response 3: The survey instrument used for data collection was had demographic (16 questions) and health care services utilization (20 questions) parts. The demographic section included questions on gender, race, marital status, monthly income and employment status, self-identified by the participants. The health care utilization portion of the survey consisted of questions about the frequency of visits to various healthcare services and support groups, rather than Likert type questions. The 10- point scale was used to determine how many times the caregiver utilized healthcare services in the past 6 months, with 0 being not using any services and 10 being utilizing such services 10 or more times in the last 6 months.

Pointe 4: Was your sample randomly selected?

Response 4: Has been mentioned in the text a few times (lines 14-15, 97-98, 223-225).

Point 5: Line 212-216 Some of your text should be in the methodology section too. 

Response 5: The above-mentioned paragraph is in the Discussion section. The purpose of this paragraph is to explain our findings, which is what the text is doing. The paragraph doesn’t contain any information on methodology: thus, the authors found Discussion section to be the appropriate place for that paragraph. 

Point 6: In my opinion, you need to expand your discussion based on your results. More comparisons with the current literature should be made. Do not focus only on your limitations. 

Response 6: The comparisons with the current literature have been made. See the citations and the references in the manuscript (lines 208, 213, 215). Additionally, see below the paragraph, which can also be found in the manuscript highlighted in yellow: “The findings in the scientific literature regarding whether the health status associated with caregivers of dementia patients is poorer or better are inconsistent. There is also inconsistency of the available literature on healthcare utilization differences among caregivers and non-caregivers. Some of the studies show poorer health status [18,19], behavioral problems [20], while others report caregiving having positive effects on health provided caregiving activities are not too heavy a burden [21], decreased hospitalizations [22] and even decreased all-cause mortality [23].”  

Reviewer 3 Report

This manuscript addresses an important issue as are the problems of caregivers of patients with dementia.

However, there is not a very in depth work.

Some other thinks that I would like to point out:

There is not a conclusion section The authors do not give the average age of the caregivers and controls. I think that this is an important data to have, first in order to determine the homogeneity of the two groups and also to know better the profile of this people. The authors found differences in the amount of medicines (prescribed and OTC), used by the caregivers. This in an interesting finding I should be further investigated. It is not due to a difference in the profile between caregivers and controls? If this is not, what are the most consumed drugs by caregivers and could they be related to their work attending these patients? (for example, more use of anxiolytics, etc)

Author Response

Point 1: Extensive English editing

Response 1: The manuscript has been checked by a native English-speaking colleague.

Point 2: there is no conclusion section.

Response 2:  The conclusion section is not a requirement per “Pharmacy” journal criteria. It is advised to include a Conclusion section only of the Discussion section is unusually long or complex, which is not the case with the present manuscript.

Point 3: The authors do not give the average age of the caregivers and controls. I think that this is an important data to have, first in order to determine the homogeneity of the two groups and also to know better the profile of this people.

Response 3: Please, see lines 162 – 163 in the updated manuscript, which reads as follows: “The mean age of the participants was homogenous between the two groups with 47 and 52 years old for the control and the caregiver group respectively”.

Point 4: The authors found differences in the amount of medicines (prescribed and OTC), used by the caregivers. This in an interesting finding I should be further investigated. It is not due to a difference in the profile between caregivers and controls? If this is not, what are the most consumed drugs by caregivers and could they be related to their work attending these patients? (for example, more use of anxiolytics, etc)

Response 4: The survey only asked about the number of prescription and non-prescription medication use, rather than the type/class of medications used. Since we do not know the type of medications consumed, we cannot identify whether the difference between the caregivers’ and the control groups was due to a difference in their profile. See the manuscript for additional elaboration on medication use by the caregivers. See lines 239-244 in the updated version, which reads as follows: The use of over-the-counter (OTC) medications and prescription medications were also significantly higher compared to that of the control group. A study by National Alliance for Caregiving and Evercare reported similar trends in terms of increased medication use in caregivers. This study reports that caregivers use prescription and psychotropic drugs more than non-caregivers. [18] The survey only identified the frequency or prescription and OTC medication use, rather than the class of medications consumed. Regardless, it can be implied that this finding was related to the physical and psychological burden associated with caregiving.”   

Round 2

Reviewer 1 Report

Ther are still areas of the manuscript which need improvement. Some are points previously raised and not addressed; some are as a result of newly added sections. My comments are in the annotated review attached.

Author Response

Reviewer 1

Point 1: referencing style is wrong.

Response 1: Please, see the revised version of the manuscript. All changes have been highlighted in yellow.

Point 2: type

Response 2: fixed in the manuscript

Point 3: The sentence is difficult to read

Response 3: the sentence now reads as follows: “Providers of dementia patients can also teach their caregivers some techniques of coping with anger, depression and frustration, and thus improve caregiver’s quality of life.”

Point 4: line 106-107 – be more specific

Response 4: The paragraph now reads as follows: “After obtaining permission from various local community centers, churches, mosques and temples in Illinois suburbs, the researchers visited the facilities on their weekly service days. The surveys were distributed to the participants of the daily services. The choice of locations was based on vicinity and convenience of obtaining approvals.”

Point 5: “ambiguity” – incorrect term

Response 5: The researchers believe ambiguity is the right word to use in order to convey the message in this sentence.

Point 6: the word “various” is overused.

Response 6: the word has been removed from that sentence. See line 132.

Point 7: capitalize SPSS

Response 7: see lines 150-151

Point 8: prefers “ethnicity”

Response 8: see table 1, lines 135, 275 highlighted in yellow.

Point 9: table 2 and figure 1 – needs legends

Response 9: A title has been added to the table. The legend was not added since the table was very self-explanatory.

Point 10: The sentence: “Since most of the caregivers were homemakers, they got to spend the majority of their time with dementia patients, which could add to their stress, physical and mental exhaustion, and adversely impact their psychological well-being” need to be referenced.

Response 10: This sentence discusses the study findings and tries to explain the reasons we found what we found. All the studies available on the caregiving impact on caregiver’s health have been referenced in paragraph above. The sentence” Without chance to rest and rejuvenate ….” Has been removed.

Point 11: [25,26,6] – references should appear in order

Response 11: Per AMA referencing guidelines, the order of the citation is per the order of their appearance in the text, rather than in numerical order. We have used an electronic/automated reference tool, which has confirmed this idea.

Point 12: Where is the data for the control group? why is this not all included in Table 1? Education not included in tables.

Response 12:

The control group’s and the caregiver’s group education information is included in the table below table 1. The researchers feel that the information need not be duplicated both in the table and the text.

Most participants were High School graduates (22%) in caregivers group and (28%) in the control group, and College graduates (68%) in the control group (60%) in the control group, with the majority making a monthly income of over 3500 dollars per month (64%) and (72%) in the control group.

Reviewer 2 Report

My comment about the paragraph in the discussion section remains the same. In my opinion, you may need to rewrite it. 

Author Response

The discussion section has been updated. Please see the attached edited manuscript. Thanks.

Reviewer 3 Report

The authors have answered the questions I made and made some improvements in the manuscript